# Assessing Temperature-Dependent Deltamethrin Toxicity in Various *kdr* Genotypes of *Aedes aegypti* Mosquitoes

**DOI:** 10.3390/insects16030254

**Published:** 2025-03-01

**Authors:** Joshua Kalmouni, Brook M. Jensen, Joshua Ain, Krijn P. Paaijmans, Silvie Huijben

**Affiliations:** 1The Center for Evolution and Medicine, School of Life Sciences, Arizona State University, Tempe, AZ 85287, USA; jkalmoun@asu.edu (J.K.); bmjense4@asu.edu (B.M.J.); jain1@asu.edu (J.A.); kpaaijma@asu.edu (K.P.P.); 2Simon A. Levin Mathematical, Computational and Modeling Sciences Center, Arizona State University, Tempe, AZ 85287, USA; 3WITS Research Institute for Malaria, Faculty of Health Sciences, University of the Witwatersrand, Johannesburg 2193, South Africa

**Keywords:** insecticide resistance, pyrethroids, temperature coefficient, climate change, mosquito, *Aedes aegypti*, voltage-gated sodium channel, vector control, knock-down resistance

## Abstract

Mosquitoes can spread deadly diseases like dengue and Zika, and insecticides are a critical tool for controlling them. However, insecticide resistance is a growing problem, and understanding how environmental factors like temperature influence resistance is essential for effective control strategies. This study focused on *Aedes aegypti* mosquitoes, testing how their genetic makeup and exposure to different temperatures affect the effectiveness of a common insecticide (deltamethrin). Mosquitoes were either fully resistant (IICC), partially resistant (VVCC), or a cross between the two (VICC), and tested at three temperatures: 22 °C, 27 °C, and 32 °C. The results showed that partially resistant female mosquitoes were less affected by the insecticide as the temperature increased, whereas this pattern was not observed in males or other genetic groups. These findings highlight the complex interplay between genetic resistance and environmental conditions, suggesting that insecticide effectiveness can vary between climate zones or locally between different weather conditions. By understanding these dynamics, we can design more effective mosquito control programs and slow the spread of insecticide resistance, ultimately improving efforts to protect communities from mosquito-borne diseases.

## 1. Introduction

Vector-borne diseases are responsible for over 700,000 deaths annually around the globe, the majority of these caused by mosquitoes [1]. Despite their significant impact, many mosquito-borne diseases lack effective drugs or vaccines for clinical treatment and prevention, necessitating heavy reliance on insecticides to control mosquito populations. Due to their widespread usage, mosquitoes worldwide have evolved resistance to many, if not all, classes of insecticides [2,3,4,5,6]. Resistance surveillance is important for the early detection of resistance and implementing resistance management strategies. Standardized insecticide susceptibility assays are employed to assess resistance prevalence across locations and over time [7,8,9]. Nonetheless, these assays, conducted under controlled conditions, may not accurately reflect practical resistance levels in the field [10,11]. Various environmental factors influence insecticide susceptibility, with ambient temperature being a key determinant [10]. Understanding the temperature–toxicity relationship is essential for interpreting assay results, as mosquitoes in field conditions will experience different temperatures than those in laboratory settings. Furthermore, regional temperature variability and climate-change-driven shifts in environmental conditions shape the selective landscape for the evolution of resistance. Thus, understanding the temperature–toxicity landscape is key for effective vector control strategies that are tailored to diverse ecological settings.

*Aedes aegypti* is a highly adaptable mosquito that originates from Africa but has spread globally, with high prevalence in the Americas, Asia, and Oceania. It is mostly an urban vector due to its preference to use artificial containers as larval habitats and its preference to feed on humans [12]. Insecticide resistance, particularly against pyrethroids, is common among *Ae. aegypti* populations and has been reported on all mosquito-inhabiting continents [13,14,15,16,17,18,19] except for Australia [17,20]. Mutations in the voltage-gated sodium channel (VGSC) are particularly common, leading to knockdown resistance (*kdr*) against DDT and pyrethroids. Several mutations are commonly found in this gene region. First, F1534C appears to have evolved at least two independent times in a process of convergent evolution [17]. Additional mutations V1016I or V1016G are found in many *Ae. aegypti* populations in combination with F1534C [15,17,21]. A recently identified *kdr* mutation in *Ae. aegypti*, V410L, provides resistance to both type I and type II pyrethroids [22]. This mutation was found to be in linkage disequilibrium with the V1016I and F1534C mutations [23]. A variety of additional mutations in the VGSC coding genes have been reported to confer resistance to pyrethroids as well (reviewed in [24,25]).

Since *Ae. aegypti* mosquitoes are widely distributed across the globe, they experience a broad range of temperatures. Their distribution is limited by certain temperature extremes, particularly with minimum annual temperatures restricting their spread to higher latitudes [26], and temperatures they encounter in different climate zones or locally throughout seasons vary considerably. For instance, *Ae. aegypti* can endure winters with temperatures around 0 °C in southern Europe [27] and survive in the scorching summers of Maricopa County, Arizona, where mean monthly temperatures reach 35 °C [28]. Indeed, *Ae. aegypti* has demonstrated the ability to survive a wide temperature range [29]. This variability is important because insecticide exposures, therefore, also occur under diverse temperature conditions, which can significantly influence the effectiveness of the insecticides, either positively or negatively.

In poikilotherms, such as mosquitoes, ambient temperature affects both organismal and chemical processes [30]. Higher temperatures are associated with increased metabolism, which enhances the uptake, detoxification, and excretion of insecticides. These temperature-dependent effects differ between compounds of different insecticide classes since elevated temperatures can alter the biotransformation of insecticides, affecting their toxicity [30,31]. For instance, the toxicity of organophosphates is dependent on the metabolic activation of the parent compound, which increases with higher temperatures due to enhanced enzymatic activity. Pyrethroids, in contrast, are inherently toxic, and increased temperatures, therefore, would lead to greater hydrolysis and the elimination of the toxic compounds. Hence, in theory, organophosphate toxicity would increase with higher temperatures (known as a positive temperature toxicity coefficient) while pyrethroid toxicity would decrease (a negative temperature toxicity coefficient). However, for type 2 pyrethroids, such as deltamethrin, increased ambient temperatures may also result in higher toxicity, likely due to the alpha-cyano group present in these compounds, though the mechanisms remain unclear [30]. Moreover, voltage-gated ion channel kinetics are shown to be influenced by temperature, which may affect the toxicity of insecticides targeting these channels, such as pyrethroids [32]. Electrical signaling can be amplified at higher temperatures, resulting in more excitable channels. This could potentially enhance the toxic effects of pyrethroids as these work by causing an overexcitation of these channels. In contrast, at lower temperatures, the number of excitable channels decreases, and toxicity would theoretically decrease [32].

While the theory provides a clear framework for how temperature might influence insecticide toxicity in mosquitoes, experimental data remain limited and reveal a more complex and nuanced picture. To enable comparisons of insecticide susceptibility across time and space, insecticide resistance surveillance in mosquitoes is typically conducted under the standardized temperature conditions of 27 ± 2 °C [8,9,33]. Consequently, limited data are available on the effects of temperature on insecticide resistance in mosquitoes. Nonetheless, various studies have examined the influence of ambient temperature, during the larval stage and/or exposure phase, on insecticide toxicity across different mosquito species and insecticides. The direction and the magnitude of the temperature–toxicity coefficients differ greatly between and within species and insecticides. Overall, the organophosphates malathion and chlorpyriphos have some evidence of increased mortality at higher temperatures [34,35,36,37]. While no studies have reported negative correlations, some have been unable to establish a clear temperature–toxicity relationship with malathion [38,39,40]. For pyrethroids, the results are much more conflicting, with positive and negative associations reported, as well as a combination of both across a temperature range [34,37,38,41,42,43,44,45,46]. Only a few studies have been performed on *Ae. aegypti* exposed to pyrethroids at different temperatures. Whiten and Peterson showed a non-linear relationship, i.e., mortality following permethrin exposure tended to be highest at the more extreme temperatures (16 °C and 32–34 °C) compared to 30 °C, though not all comparisons were significantly different [45]. Salinas and colleagues kept *Ae. aegypti* and *Ae. albopictus* from pupae onwards at three daily temperature fluctuation regimes mimicking temperatures in Rio Grande Valley (McAllen, TX, USA) in the summer, spring/fall, and winter. They found opposing temperature–toxicity relationships for *Ae. albopictus* mosquitoes exposed to deltamethrin and permethrin: permethrin toxicity decreased at higher temperatures, whereas deltamethrin toxicity increased at higher temperatures [46]. *Ae. aegypti* mosquitoes exposed to permethrin had, in contrast to *Ae. albopictus*, higher mortality at higher temperatures, and no relationship was found between temperature and toxicity for deltamethrin [46]. A major limitation of these studies is that temperature effects are often assessed on a single day and/or with the same mosquito generation, resulting in pseudoreplication [47]. This raises the possibility that conflicting results may stem from experimental noise rather than true biological variation, highlighting the need for more robust experimental designs. Additionally, most studies use a single insecticide dose, typically diagnostic, which may lack the sensitivity to detect temperature effects, especially in highly resistant or fully susceptible strains. Utilizing a range of dosages tested on multiple mosquito batches and calculating the lethal concentration killing 50% of mosquitoes (LC50), as performed by e.g., Whiten and Peterson [45], provides a more reliable approach to addressing these complex questions.

A key remaining question is how the relationship between temperature and toxicity interacts with insecticide resistance mechanisms in mosquitoes. Theoretically, temperature could impact resistant and susceptible mosquitoes differently. With increasing metabolic rates and thus increased insecticide degradation at higher temperatures, resistance mechanisms may be particularly effective in vector populations where metabolic resistance depends on temperature-sensitive mechanisms. Mutations in the VGSC in the fruit fly *Drosophila melanogaster* have been used historically due to their temperature-sensitive phenotype leading to reversible paralysis at higher temperatures [48]. Therefore, polymorphisms in this gene could also lead to differential temperature–toxicity relationships. Supporting this, a study in China found that the *kdr* 1534 mutation frequency was positively correlated with the annual average temperature [49]. However, correlation does not imply causation. A few studies have experimentally examined the effect of temperature during exposure on both resistant and susceptible mosquitoes of the same species. Notably, Glunt et al. [43] and Hodjati and Curtis [42] conducted such studies using *Anopheles* species. Although these studies found slightly different patterns, the use of a single dose for both resistant and susceptible strains and the lack of true replicates indicate that more research is needed to confirm these as true biological effects. In this study, we investigated how temperature influences the toxicity of the type II pyrethroid deltamethrin on *Ae. aegypti* mosquitoes across three *kdr* genotypes using LC50 measurements and true replicates.

## 2. Materials and Methods

### 2.1. Mosquito Genotypes and Rearing Procedures

In this study, we assessed *Aedes aegypti* mosquitoes of three genotypes: VVCC, IICC, and VICC. All genotypes carried two copies of the F1534C mutation. The VVCC genotype had two wildtype alleles at the 1016 locus of the *kdr* gene (1016V), IICC had two mutant alleles (V1016I), and VICC was heterozygous at this locus. The genotypes VVCC and IICC were established in the lab from field collections in St. Augustine, Florida, in June 2016 [50]. To obtain heterozygote mosquitoes for this study, male VVCC mosquitoes were crossed with virgin female IICC mosquitoes. In brief, virgin IICC females were ensured by isolating pupae individually in tubes until eclosion, after which they were housed with VVCC males. The resulting eggs were reared to produce the VICC genotype. This process was repeated across four separate crosses from different parent generations. To rear adult mosquitoes, eggs were submerged in deionized (DI) water and placed in a 0.2 cu. ft vacuum desiccator (Bel-Art – SP Scienceware, Wayne, NJ, USA) for 25–30 min. Hatched larvae were placed in plastic containers (17 × 31.5 × 9 cm) filled halfway with DI water, with approximately 400 larvae per container. The larvae were reared in environmental chambers (Darwin Chambers Company, St. Louis, MO, USA) maintained at 27 °C, 75% relative humidity, and a 12:12 light:dark cycle. Larval trays were refreshed with clean DI water and fed ground Cichlid fish food pellets (Aqueon, Franklin, WI, USA) every other day. Adult mosquitoes were provided with a 10% sucrose solution delivered through a cotton wool ball, which was replaced every other day.

### 2.2. Climate Boxes

Insulated shipping containers made from Bio Foam (Polar Tech, Genoa, IL, USA, 48.3 × 30.5 × 40.6 cm, inner dimensions) were used to create climate-controlled boxes [51]. Temperature was maintained in these climate boxes using reptile heat cables (Zoo Med, San Luis Obispo, CA, USA, Model RHC15), and humidity was maintained using a terrarium humidifier (Coospider, Levallois-Perret, France, Model 15hf98-4h287). Climate boxes were set at 75% RH and a temperature of either 22 °C, 27 °C, or 32 °C. Temperature and humidity were regulated within the climate box with a temperature and humidity controller (Digiten, via Amazon, Seattle, WA, USA, model: DHTC-1011). The climate box set at 22 °C was placed in a standalone incubator (BioCold Environmental, Ellisville, MO, USA, Model BC26-IN) set at 20 °C to achieve a temperature lower than the ambient room temperature. A data logger (Omega Engineering Inc., Michigan City, IN, USA, Model OM-92) was placed into each box, and temperature/humidity data were logged. Climate boxes were kept on a 12:12 h L:D cycle using 4W submersible LED aquarium lights (MingDak, via Amazon, Seattle, WA, USA) and an analog timer.

### 2.3. Paper Impregnation

Insecticide papers were prepared with deltamethrin as the insecticide, analytical grade olive oil as the carrier oil, and acetone as the solvent, following WHO standard operating procedures for paper impregnation, with some minor modifications [52]. Paper concentrations ranged from 0.001% to 7%. In brief, serial dilutions of deltamethrin in acetone were performed to obtain the desired concentration of insecticide in 2 mL of acetone per paper (Appendix A). The resulting solution was then combined with 0.71 mL of olive oil, thoroughly mixed by inversion, and vortexed for at least 10 s. All solutions were prepared gravimetrically using a precision balance (Sartorius, Göttingen, Germany, Model Quintix). The resulting mixture was evenly distributed onto pieces of 12 × 15 cm Whatman filter paper by slowly discharging the insecticide solution in a grid using a micropipette. Papers were subsequently placed onto a drying rack in a dark fume hood for 24 h. After the drying period, papers were covered in tin foil to eliminate light degradation and stored at 4 °C. Per WHO guidelines, impregnated papers were used a maximum of six times and stored at 4 °C between assays. Deltamethrin concentrations were selected through range-finding pilot experiments to determine at least five concentrations for each genotype, resulting in mortality rates greater than 0% but less than 100%.

### 2.4. WHO Tube Test

Two-to-five-day-old mosquitoes were acclimatized to one of the three temperature conditions for 24 h prior to the initiation of the WHO tube assay. This acclimation period was chosen based on previous research indicating that thermal acclimation in insects is typically complete within 20 h [53]. To acclimatize mosquitoes, mosquitoes were separated by sex into a smaller, secondary cage (a 2 L plastic container with netting on top) of approximately 100 to 150 individuals. This cage was then placed inside the climate box with access to a 10% sucrose solution. The next day, mosquitoes were exposed to deltamethrin-impregnated or control papers following standard WHO tube test methodology [8]. In brief, 15–30 mosquitoes—depending on availability—were aspirated into holding tubes lined with untreated filter papers 23 h after initial acclimatization and placed back in their climate boxes for one hour. Next, mosquitoes were transferred to a tube lined with an insecticide-impregnated or control paper coated with the carrier oil and placed back in their climate box. Different insecticide concentrations were randomized by tube and given a blind ID to prevent bias in mortality assessments. One untreated tube containing mosquitoes of the same sex, genotype, and temperature was used as a control for each assay with a maximum of five exposure tubes per control tube. After one hour of exposure, mosquitoes were transferred back into the holding tubes, returned to the climate boxes, and given a small piece of cotton soaked in a 10% sucrose solution on top of the tube’s netting. Mortality was assessed 24 h after the exposure ended. Knocked-down and moribund mosquitoes were counted as dead. Assays in which the control mortality exceeded 20% were repeated. Mortality observations for each combination of temperature, genotype, and sex were collected from a minimum of two independent mosquito batches and generations. For each combination, at least four observations were recorded to construct the dose–response curve and calculate the LC50, with the number of observations ranging from 4 to 15 (mean: 7.2). All assays were performed randomly and conducted over an 18-month period.

### 2.5. Statistical Analysis

The Abbott’s correction was performed when mortality in the control tube exceeded 5%. A generalized linear model (GLM) was used to analyze the mortality data as a function of the logarithm (base 10) of concentration and temperature for each genotype of each sex. An interaction term was fitted between concentration and temperature and subsequently removed if insignificant. The GLM utilized a quasibinomial family with a probit link function to account for the overdispersion of the mortality data. Furthermore, GLMs were executed across all aggregated temperatures to estimate the concentration at which 50% mortality occurred (LC50) for each sex and genotype [54]. The 95% confidence intervals (95% CI) were calculated by multiplying the standard error of the LC50 by 1.96. LC50 values were deemed significantly different if their 95% CI did not overlap. All analyses were performed in R version 4.3.2 [55].

## 3. Results

A total of 241 WHO tubes were run across 63 separate assays (Table 1), assessing a total of 4897 mosquitoes. Variations in mosquito availability among different genotypes led to uneven sample sizes, with particularly larger groups observed for VICC at 22 °C and 32 °C compared to other genotype and temperature combinations.

Our assays confirmed the role of the V1016I mutation on deltamethrin susceptibility, with VVCC being the most susceptible, followed by VICC. Mosquitoes with the IICC genotype were the least susceptible to deltamethrin (Figure 1). The calculated LC50, pooled across temperatures, of VVCC mosquitoes was 0.029% (0.025–0.033) for females and 0.017% (0.013–0.021) for males. The LC50s for VICC mosquitoes were 0.090% (0.080–0.101) for females and 0.044% (0.037–0.051) for males. The LC50s for IICC mosquitoes were 0.45% (0.38–0.54) for females and 0.36% (0.30–0.42) for males. Thus, IICC mosquitoes were 15.5 times less susceptible than VVCC in the females and 21.2 times less susceptible in the males. The heterozygous (VICC) mosquitoes were considerably more susceptible than the homozygous mutant (IICC) mosquitoes but significantly less susceptible than the homozygous wildtype (VVCC) mosquitoes (3.1 times for females and 2.6 times for males). Females of the genotypes VVCC and VICC were significantly less susceptible to insecticides than males. No significant differences were observed between males and females of the genotype IICC (Figure 2).

For females of the most susceptible genotype (VVCC), higher temperatures were strongly associated with a lower level of susceptibility to the insecticide (Temperature: t(18) = 8.46, *p* < 0.001, Figure 3A, Table 2). However, this association was not observed for VVCC males (Figure 3B) or any of the other genotypes (*p* > 0.05, Figure 3C–F, Table 2).

## 4. Discussion

Our findings demonstrate the influence of the V1016I allele on deltamethrin susceptibility. Combined with the F1534C mutation, mosquitoes with two copies of the V1016I mutant allele were significantly more resistant to deltamethrin than mosquitoes with only one or zero copies, regardless of sex. However, overall, males were consistently more susceptible than females, a well-documented phenomenon possibly due to their smaller body size [56]. A decrease in pyrethroid toxicity with increasing temperature was found exclusively in female mosquitoes with the homozygous wildtype at the 1016 locus (VVCC), indicating a negative temperature coefficient in this group of mosquitoes. However, no temperature–toxicity relationship was observed in VVCC males or in the other genotypes of either sex. These findings suggest that temperature may differentially influence pyrethroid toxicity depending on specific *kdr* mutations, highlighting the potential complexity of temperature–genotype interactions in determining insecticide susceptibility. Interestingly, the least resistant female mosquitoes (VVCC) at warmer temperatures exhibited increased resistance and were phenotypically nearing the susceptibility of heterozygous females (VICC). This would suggest that the selection for the V1016I mutation, and potentially other target-site resistance mutations, might be reduced in warmer climates and increased in colder climates.

Metabolic rate, along with most biological activity, is temperature-dependent, which tends to increase with temperature within the range of the normal activity of an organism [53,57,58,59]. However, the rates of such biological processes are not always predictably elevated with temperature and are influenced by complex ecological and evolutionary factors [59,60]. Negative temperature coefficients have been documented for pyrethroids in various arthropods, often attributed to decreased metabolic activity at lower temperatures impeding the breakdown and elimination of toxins [30,61]. Additionally, lower temperatures may reduce electrical signaling and channel excitability, potentially altering insecticide efficacy. Lastly, colder temperatures reduce biotransformation (i.e., chemical modification), leading to greater accumulation of toxic parent pyrethroids, which are more harmful than their metabolites [62]. While a negative temperature coefficient is more commonly associated with type I pyrethroids [30,42], our study using deltamethrin, a type II pyrethroid, also revealed this pattern. However, this effect was only seen in female VVCC mosquitoes. It is unclear why this association is only seen in this particular group of mosquitoes. Few studies have assessed the temperature coefficient for pyrethroids in both resistant and susceptible mosquitoes. In one study, researchers found a negative coefficient for susceptible *An. arabiensis* mosquitoes tested at 18 °C, 25 °C, and 30 °C, while the resistant strain showed a non-linear response with higher mortality at both lower and higher than optimum temperatures. In the same study, resistant and susceptible *An. funestus* were also tested with different patterns found in both [43]. However, in both cases, the experiment was not repeated over multiple mosquito batches, leaving the possibility that the results were influenced by factors such as rearing condition variability, generational differences, paper impregnation consistency, experimenter technique, or ambient climatic conditions [43]. In our study, we conducted replicates independently over a period of 18 months and across multiple generations to reduce the chance of false positives through pseudoreplication [47].

It is unclear why such temperature–toxicity observations would only be observed in VVCC mosquitoes and not in the more resistant genotypes. VICC and IICC genotypes have reduced susceptibility associated with the synergistic effect of the V1016I and F1534C mutations [63]. One possible explanation is that the higher baseline resistance of the VICC and IICC genotypes may obscure or mitigate temperature-dependent effects caused by changes in metabolic rate and/or voltage-gated ion channel kinetics. In resistant mosquitoes, the dose–response curve is shifted toward higher concentrations. Small changes at lower concentrations are more discernible on a logarithmic scale than at higher concentrations, where differences become compressed. As a result, subtle changes in toxicity due to temperature effects may be less apparent at the elevated dosages required to affect resistant mosquitoes, effectively masking the temperature–toxicity relationship [64]. In contrast, the least resistant mosquitoes (VVCC) were affected at lower insecticide concentrations, where we hypothesize that changes in toxicity due to temperature are more likely to be detectable. Additionally, the temperature range tested in this study (10 degrees, from 22 °C to 32 °C) may have been too narrow to capture the full spectrum of temperature–toxicity interactions. For example, Hodjati and Curtis examined temperatures across a 21 °C range (16–37 °C) [42], Glunt et al. used a 12 °C range (18–30 °C) [43], and Kalmouni et al. tested a 20 °C range (15–35 °C) [37]. Broader temperature ranges might reveal non-linear or genotype-specific responses, as observed in previous studies of other mosquito species [43]. However, these explanations do not account for the absence of a similar pattern in male VVCC mosquitoes. A strong interaction between sex, temperature, and resistance genotype on insecticide toxicity was also found in a *Drosophila melanogaster* model system [64]. These combined findings highlight the complex interplay between temperature, insecticide toxicity, genetic variation, and sex. This complexity makes predicting temperature–toxicity relationships for different insecticides and resistance mechanisms across mosquito vectors difficult. Ultimately, this study represents an initial step toward unraveling these interactions, and further research is needed to draw broader conclusions across mosquito species, genotypes, and insecticides.

There are several limitations to our study design that may impact the generalizability and applicability of the findings. First, the *Ae. aegypti* mosquitoes used in the experiments were lab-adapted strains, isolated from a single site, and housed under lab conditions for 16–20 generations. Lab strains are commonly used for these types of studies. However, since they have been reared under optimal conditions at 27 °C, they may have adapted to these conditions. Lab-colonized mosquitoes have been shown to have greater nutritional reserves [65] and exhibit lower genetic and transcriptomic diversity [66,67] than field populations. *Ae. aegypti* from climatically diverse locations in Mexico did indeed show natural variation in thermal tolerance and adaptation to laboratory conditions within ten generations [68]. This adaptation could potentially impact the temperature–toxicity association, and it remains to be studied whether temperature coefficients are different between natural populations and lab-adapted populations. Second, the insecticide exposure in this study was conducted using WHO tube tests, which involve tarsal exposure to the insecticides. The purpose of the WHO tube assay is to determine the presence of insecticide resistance in field populations and to provide a standardized approach for assessing insecticide susceptibility. However, it does not fully replicate the natural route of exposure. In practical applications, fogging is often used for *Ae. aegypti* control. The dynamics of insecticide droplets, influenced by temperature and other environmental factors, are not captured in the WHO tube tests. These dynamics likely play a significant role in the effectiveness of insecticides under field conditions but fall outside the scope of this study. Therefore, the results of this study may not fully represent the efficacy of insecticides when applied via fogging in various environmental conditions. Conversely, our results show the importance of incorporating ambient temperatures or allowing lab-reared mosquitoes to acclimatize when interpreting results from semi-field trials with ultra-low volume space sprays (e.g., [69,70]). In this study, mosquitoes were acclimated for 24 h, an intermediate duration compared to other temperature–toxicity studies that acclimated mosquitoes for one hour (e.g., [37]) or multiple days [34]. A 24 h period likely minimized temperature change stress, but it also could allow for increased heat shock protein production at 32 °C, which may have contributed to reduced mortality [71]. Lastly, our study focused solely on the V1016I mutation in the VGSC with a constant F1534C mutation background, both involved in pyrethroid resistance [72]. Understanding how temperature coefficients depend on various insecticide resistance mutations is crucial, especially given that we found different responses depending on the three genotypes tested in this study. Moreover, we did not test for metabolic resistance, another important insecticide resistance mechanism. Given that temperature impacts metabolic rates, there could be significant interactions between temperature and metabolic resistance that were not explored in this study. Further research is needed to investigate these variables and validate the findings under more diverse and realistic settings.

## 5. Conclusions

This study presents, for the first time, data on the influence of the V1016I mutation on deltamethrin susceptibility in *Ae. aegypti* mosquitoes, in combination with the F1534C *kdr* mutation, across different temperatures. These findings highlight the complex interactions between genetic mutations and environmental factors, such as temperature, in determining insecticide resistance. Given that *Ae. aegypti* is one of the most important vectors of mosquito-borne diseases and inhabits a wide range of climate conditions, it is crucial to understand not only the efficacy of our interventions under local climate conditions but also how these conditions might shape selective pressures for resistance evolution. Our data on these *Ae. aegypti* genotypes suggest that a single temperature coefficient may not exist. Instead, these interactions are likely complex, involving specific loci, different resistance mechanisms, genetic backgrounds, species, and insecticides. Furthermore, extreme temperature exposures such as 37 and 39 °C during different life stages could impact resistance [38]. Repeating studies like this with natural populations instead of lab-reared mosquitoes would provide valuable insights. In particular, comparing populations from different climate regions, where mosquitoes have experienced varying natural temperatures, could reveal important ecological and evolutionary differences. As we continue to unravel these interactions and their implications, collaboration and integration of findings from diverse studies is essential. By doing so, we can develop more evidence-based intervention strategies based on local biotic and abiotic conditions aimed at helping us move closer to the goal of reducing the burden of vector-borne diseases worldwide.

## Figures and Tables

**Figure 1 insects-16-00254-f001:**
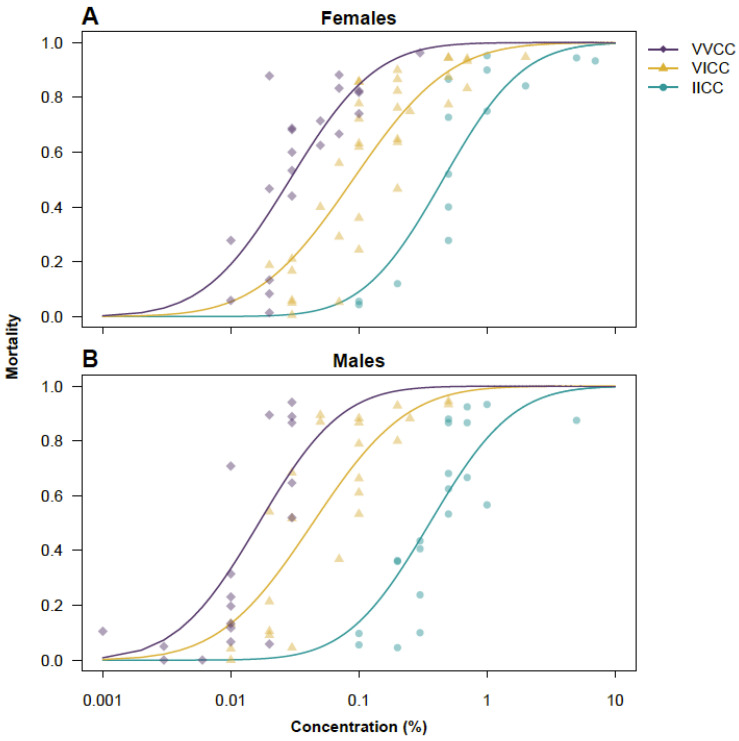
Dose–response curves of female (**A**) and male (**B**) *Aedes aegypti* mosquitoes of *kdr* genotype VVCC (purple), VICC (yellow), and IICC (cyan) exposed to deltamethrin in WHO tubes. Diamonds (VVCC), triangles (VICC), and circles (IICC) represent mortality within one tube. Trend lines are predicted mortality based on probit link regression analysis. Data by genotype are pooled across all temperatures.

**Figure 2 insects-16-00254-f002:**
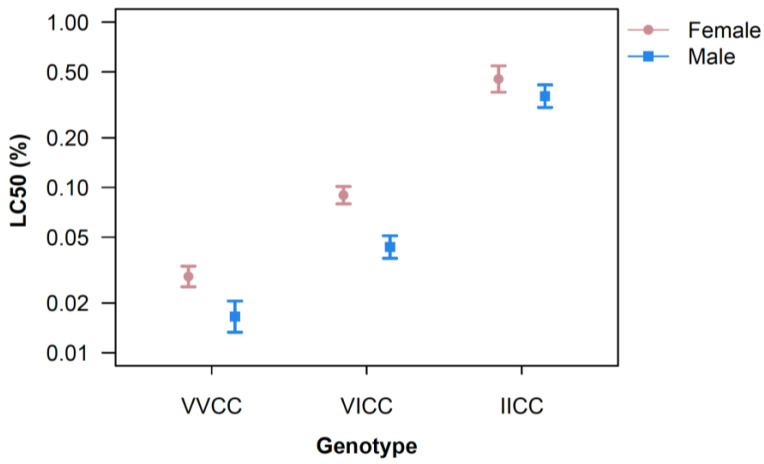
Lethal concentration causing 50% mortality (LC50) on a probit scale by genotype for females (pink circles) and males (blue squares). Error bars are 95% confidence intervals.

**Figure 3 insects-16-00254-f003:**
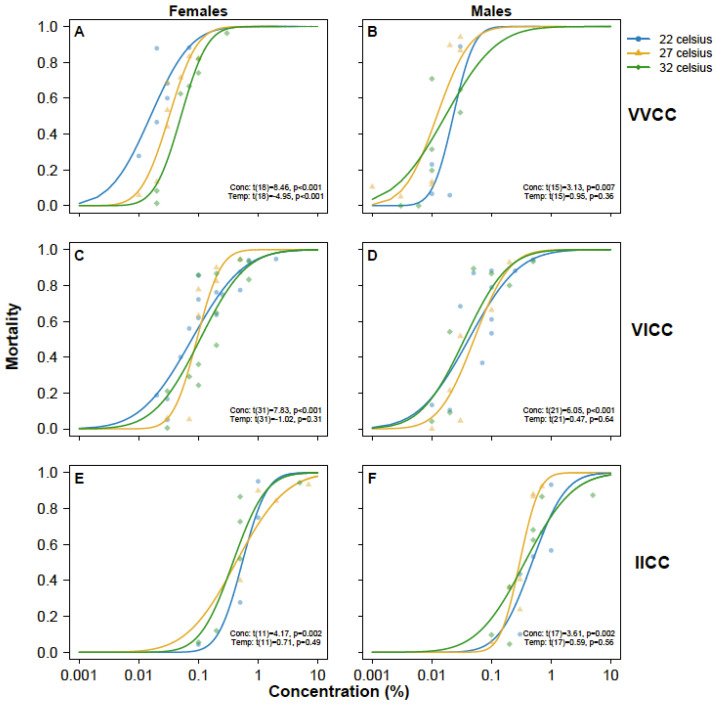
Dose–response curves of female (left column) and male (right column) *Aedes aegypti* mosquitoes of *kdr* genotype VVCC (**A**,**B**), VICC (**C**,**D**), and IICC (**E**,**F**) exposed to deltamethrin in WHO tubes at 22 °C (blue lines), 27 °C (orange lines), and 32 °C (green lines). Each symbol (22 °C: circle; 27 °C: triangle; 32 °C: diamond) represents mortality within one tube. Trend lines are predicted mortality based on probit link regression analysis. Results from regression analysis for the effect of concentration (Conc) and temperature (Temp) are provided in the bottom-right corner of each panel.

**Table 1 insects-16-00254-t001:** WHO paper concentrations (Conc.) in percentage used for each genotype, temperature (°C), and sex combination. The column ‘n’ indicates the number of tubes tested for each concentration. A minimum of two assays (including one or more concentrations plus a control tube) were performed for each genotype, temperature, and sex combination. Each assay within the same genotype, temperature, and sex combination was performed with a different mosquito batch. Bolded concentrations denote data that were used for LC50 calculations (mortality was greater than 0 and less than 100%).

	VVCC				VICC				IICC			
	♀		♂		♀		♂		♀		♂	
	Conc.	n	Conc.	n	Conc.	n	Conc.	n	Conc.	n	Conc.	n
22 °C	control	5	control	2	control	5	control	5	control	5	control	3
	0.005	2	0.003	1	**0.02**	**2**	**0.01**	**2 ***	**0.1**	**1**	0.1	1
	**0.01**	**2 ***	0.006	1	**0.03**	**2**	**0.02**	**2 ***	0.2	1	**0.3**	**2**
	**0.02**	**2**	**0.01**	**2**	**0.05**	**1**	**0.03**	**2 ***	**0.5**	**1**	**0.5**	**1**
	**0.03**	**3 ****	**0.02**	**1**	**0.07**	**1**	**0.05**	**1**	**1**	**2**	**0.7**	**1**
	0.05	1	**0.03**	**1**	**0.1**	**3**	**0.07**	**1**	2	1	**1**	**3 ****
	**0.07**	**1**	0.1	1	**0.2**	**3 ***	**0.1**	**4**	3	1	2	1
					**0.25**	**1**	**0.2**	**1**	4	1		
					0.3	1	**0.25**	**1**	7	1		
					**0.5**	**2**	**0.5**	**2**				
					**0.7**	**1**						
					1	1						
					**2**	**1**						
27 °C	control	2	control	3	control	3	control	3	control	4	control	3
	0.006	1	**0.001**	**1**	**0.03**	**1**	**0.01**	**1**	0.1	1	**0.1**	**1**
	**0.01**	**1**	**0.003**	**1**	**0.07**	**1**	**0.02**	**1**	0.2	1	**0.2**	**1**
	**0.02**	**1**	**0.01**	**2**	**0.1**	**2**	**0.03**	**2**	**0.5**	**1**	**0.3**	**2**
	**0.03**	**3**	**0.02**	**1**	**0.2**	**2**	**0.1**	**1**	**1**	**1**	**0.5**	**2**
	**0.05**	**1**	**0.03**	**2**	**0.5**	**2 ***	**0.2**	**1**	**2**	**2 ***	**0.7**	**1**
	**0.07**	**1**	0.1	1			0.5	1	5	1	1	1
	**0.1**	**1**							**7**	**1**	1.5	1
											2	1
32 °C	control	4	control	3	control	5	control	2	control	3	control	3
	0.006	1	**0.003**	**1**	0.01	1	**0.01**	**1**	**0.1**	**1**	0.03	1
	0.01	1	**0.006**	**1**	**0.03**	**2**	**0.02**	**2**	**0.2**	**1**	**0.1**	**1**
	**0.02**	**2**	**0.01**	**3**	**0.07**	**1**	0.03	1	**0.5**	**3**	**0.2**	**2**
	**0.05**	**1**	**0.03**	**2**	**0.1**	**3**	**0.05**	**1**	**5**	**2 ***	**0.5**	**3 ****
	**0.07**	**1**	0.3	1	**0.2**	**3**	**0.1**	**1**	7	1	**0.7**	**1**
	**0.1**	**2**			**0.5**	**3 ***	**0.2**	**1**			**5**	**1**
	**0.3**	**1**			**0.7**	**2**	**0.5**	**1**				

* Denotes only one of the replicates was included in the LC50 calculations (due to other(s) being either 0% or 100% mortality). ** denotes only 2 of the replicates were included in the LC50 calculations. ♀ denotes female mosquitoes tested; ♂ denotes male mosquitoes tested.

**Table 2 insects-16-00254-t002:** Calculated LC_50_ values with their predicted 95% CI for females and males of all three genotypes at 22 °C, 27 °C, and 32 °C.

		LC_50_ (95% Confidence Interval)	
Genotype	Temperature (°C)	Females	Males
VVCC	22	0.016 (0.012–0.021)	0.023 (0.016–0.033)
	27	0.032 (0.029–0.036)	0.012 (0.008–0.018)
	32	0.051 (0.047–0.055)	0.017 (0.012–0.023)
VICC	22	0.077 (0.065–0.092)	0.042 (0.031–0.056)
	27	0.094 (0.078–0.112)	0.051 (0.040–0.066)
	32	0.106 (0.086–0.312)	0.035 (0.024–0.050)
IICC	22	0.543 (0.413–0.715)	0.479 (0.370–0.619)
	27	0.426 (0.217–0.837)	0.297 (0.269–0.328)
	32	0.378 (0.280–0.512)	0.361 (0.273–0.478)

## Data Availability

The raw data supporting the conclusions of this article will be made available by the authors upon request.

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
