# Peer review of "Assessing Temperature-Dependent Deltamethrin Toxicity in Various kdr Genotypes of Aedes aegypti Mosquitoes"

_insects, 2025, doi:10.3390/insects16030254_

Round 1
Reviewer 1 Report
Comments and Suggestions for Authors
Manuscript submitted to insects, titled ‘Assessing temperature-dependent deltamethrin toxicity in various kdr genotypes of Aedes aegypti mosquitoes’ (insects-3408175) has reviewed.
The Authors conducted WHO tube bioassay using different doses of deltamethrin treated filter papers against pyrethroid-susceptible and -resistant Aedes aegypti acclimated under various temperature (22°C, 27°C, and 32°C) to compare lethal concentrations separately for males and females.
They found negative temperature coefficient from the susceptible female Ae. aegypti that the mosquito’s mortality was decreased as temperature increased. However, this phenomenon was not observed by the resistant populations.
Overall, the current manuscript still needs to fill in missing information in materials and methods, results, and discussion to support their findings. Please refer to the comments line by line below,
Materials and methods
L219: Please provide actual amount of deltamethrin applied using a micropipette.
L224: Please provide actual concentration tested.
L229: Need to clarify the time periods for acclimatization under three temperatures (e.g., Why 24 hours? Add related references (if any) and explain in discussion).
L230: Please provide the sizes of the secondary cage.
L247: Please provide testing doses of deltamethrin to calculate the LC50 for each strain.
L249: Please clarify whether the testing mosquitoes for each concentration used from same batch or not.
Results
Table 1: Please add legends for VVCC, VICC, IICC. Please provide detailed information about ‘Number of tubes assess’ (e.g., WHO tube bioassay, deltamethrin concentrations, number of testing mosquitoes, etc.)
L276-280: Please add more detailed information about the LC50 values obtained from which mosquitoes (acclimated under what temperature?). In addition, please describe the results from all the mosquitoes from different temperatures for the comparison with statistical information.
Figure 1: Can not find the symbols in the line graph. In addition, there is no comparison among mosquitoes from different temperatures. Please re-consider the figure to cover your experimental data without missing information.
Figure 2: It is recommended to use boxplot with statistical information for comparison between male and female, and also among the mosquito strains.
Figure 3: The dose-response curves tell trends, but it is difficult to see significance. Please add statistical information.
L300-303: Please explain in detail with actual values with statistical information.
Discussion
L314-330: Confused with duplicated sentences. Please make it simple and focus on your findings.
L334-336: Please add references related to insect metabolism under lower temperatures (22°C and below).
L336-337: Please add references related to the negative temperature coefficient with type 1 pyrethroids.
L338-348: Please add more references previously tested relations between temperature and insect metabolism influencing insecticide efficacy.
L343: Please provide more information about the temperature ranges in actual values.
L345: Please add more factors that result in variations from the WHO tube bioassay.
L346-348: Please add references to support the multiple generations affecting the bioassay testing results along with listing potential factors of variation that influencing the accuracy of the tests.
L350-373: Provide more evidence of the relation between insect mutation and sensitivity on pesticide resistance.
P364: Glunt et al., 2018 -> number?
P359-361: Add more references to explain difference between susceptible and resistance on temperature.
P361-362: Add references to provide temperature ranges in previous studies.
P376-382: Add references to explain biological differences between laboratory strain and field populations. In addition, explain why authors tested the susceptible strain that has no pesticide mutations in this study. Recommend focusing on the influence of mutations, and findings from this study.
P382-285: Please explain that the purpose of WHO tube bioassay for susceptibility test to figure out resistance of the field population, so that it has limited aspects.
P385-388: Add more information about the existing outdoor fogging efficacy bioassays. Recommended to consider the controlled temperature exposure time for testing mosquitoes as one of the variation factors (e.g., 24 hours acclimate or more?).
P392-394: Need to explain more information by adding more references related to insect mutations and pesticide resistance.
L412: Please provide actual value of the extreme temperature, and compare temperature exposure methods (e.g., 24 h acclimatize for adults vs different life stages with different exposure time, etc.)
Reviewer 2 Report
Comments and Suggestions for Authors
The manuscript describes in a relevant way the importance of monitoring insecticide resistance considering environmental parameters, particularly the temperature at which mosquito populations are breeding and on which control strategies will be implemented. However, the following aspects must be considered:
Authors mention that mosquitoes of the three genotypes used in this study were established from a field population in 2016, however, they didn’t include information about their breeding method to secure that laboratory reproduction did not affect the allelic frequencies. For example, inbreeding, particularly of the VICC genotype individuals.
It is better to describe results as more or less susceptible in your comparisons, since according to the bioassays carried out it is not possible to determine resistance as there is no susceptible reference strain (wildtype for both mutations).
It would be interesting to know the CL50 values ​​obtained in each experimental group with their confidence limits to determine significant differences between them.
The study shows and even mentions some of its limitations, such as the fact that they worked with laboratory-bred populations that were simply subjected to biological tests at different temperatures. It would be more informative, according to the purpose of the study, to evaluate resistance in populations that are naturally being bred under variable temperature conditions.

Round 2
Reviewer 1 Report
Comments and Suggestions for Authors
The Authors updated all the comments accordingly, now the manuscrip is ready to publish. Thank you very much for your hardwork.